# Investigation of Metal-Organic Framework-5 (MOF-5) as an Antitumor Drug Oridonin Sustained Release Carrier

**DOI:** 10.3390/molecules24183369

**Published:** 2019-09-16

**Authors:** Gongsen Chen, Juyuan Luo, Mengru Cai, Liuying Qin, Yibo Wang, Lili Gao, Pingqing Huang, Yingchao Yu, Yangming Ding, Xiaoxv Dong, Xingbin Yin, Jian Ni

**Affiliations:** 1School of Chinese materia medica, Beijing University of Chinese Medicine, Beijing 102488, China; gongsenchen@163.com (G.C.); luojy0329@163.com (J.L.); cmr199711@163.com (M.C.); qlytcm@163.com (L.Q.); m13164227164@163.com (Y.W.); gll_17@163.com (L.G.); fengxw0802@163.com (P.H.); yyc_zwq@163.com (Y.Y.); dingyangmingbucm@gmail.com (Y.D.); dxiaoxv@163.com (X.D.); 2Beijing Research Institute of Chinese Medicine, Beijing University of Chinese Medicine, Beijing 100029, China

**Keywords:** MOF-5, Oridonin, Antitumor, HepG2 cells, sustained release

## Abstract

Oridonin (ORI) is a natural active ingredient with strong anticancer activity. But its clinical use is restricted due to its poor water solubility, short half-life, and low bioavailability. The aim of this study is to utilize the metal organic framework material MOF-5 to load ORI in order to improve its release characteristics and bioavailability. Herein, MOF-5 was synthesized by the solvothermal method and direct addition method, and characterized by Scanning Electron Microscopy (SEM), X-Ray Diffraction (XRD), Fourier Transform Infrared Spectrometer (FTIR), Thermogravimetric Analysis (TG), Brunauer–Emmett–Teller (BET), and Dynamic Light Scattering (DLS), respectively. MOF-5 prepared by the optimal synthesis method was selected for drug-loading and in vitro release experiments. HepG2 cells were model cells. MTT assay, 4′,6-diamidino-2-phenylindole (DAPI) staining and Annexin V/PI assay were used to detect the biological safety of blank carriers and the anticancer activity of drug-loaded materials. The results showed that nano-MOF-5 prepared by the direct addition method had complete structure, uniform size and good biocompatibility, and was suitable as an ORI carrier. The drug loading of ORI@MOF-5 was 52.86% ± 0.59%. The sustained release effect was reliable, and the cumulative release rate was about 87% in 60 h. ORI@MOF-5 had significant cytotoxicity (IC50:22.99 μg/mL) and apoptosis effect on HepG2 cells. ORI@MOF-5 is hopeful to become a new anticancer sustained release preparation. MOF-5 has significant potential as a drug carrier material.

## 1. Introduction

It is expected that cancer will be the main cause of death and the most important obstacle to increasing life expectancy in every country in the world in the twenty-first century [1]. In 2018, there were 18.1 million incident cancer cases and 9.6 million cancer deaths worldwide. Cancer cases increased by 29.5% between 2006 and 2018. The incidence and mortality rates for lung cancer, breast cancer, prostate cancer, colorectal cancer, stomach cancer, and liver cancer are among the highest [2]. At present, cancer treatment methods are relatively simple and have large side effects. Therefore, it is very necessary to study new anti-tumor active ingredients and drug delivery systems to improve the efficacy and reduce the side effects of cancer treatment.

Oridonin (ORI) is an active diterpenoid extracted from the Chinese medicinal herb *Rabdosia rubescens* [3,4] that exhibits potential anticancer, antiinflammation, and antibacterial effects [5,6]. Its chemical structure is shown in Figure 1. It has been noted that ORI has antiproliferative and apoptosis-inducing effects on various types of human cancer cells, including lung, liver, colon, pancreatic, and breast cancer [7,8]. Studies showed that ORI significantly augmented JQ1-triggered apoptosis in hepatocellular carcinoma (HCC) cells and in HCC cancer stem-like cells. The enhancement of apoptosis by ORI was associated with the mitochondrial apoptosis pathway [9]. However, the great potential for clinical application of ORI is severely limited by its poor aqueous solubility and low bioavailability [10,11,12]. A variety of strategies such as graphene oxide or nanosuspensions have been attempted to deliver ORI [13,14,15], but drug loading is generally low, which limits clinical use. And less attention is paid to the effects of release characteristics. Therefore, it is of profound significance to find a suitable drug carrier for ORI to promote its clinical application.

In recent years, metal organic frameworks (MOFs) have received extensive attention in the application of drug carriers [16,17]. MOFs are highly structured, infinite network structural polymers composed of metal ions and organic ligands [18,19,20]. As drug carriers, they have the advantages of high pore surface area, adjustable pore diameter, high drug loading, long-term sustained release, good biocompatibility, and complete degradation in vivo [21,22,23,24]. Recently, Ke J. and co-workers designed and synthesized a new nitrogen heterocycle organic ligand (2E,2 E)-3,3 -(quinoline-5,8-diyl) diacrylic acid and used it to construct its first Zr-based MOF (termed as ZJU-802). ZJU-802 has been shown to have high anionic drug-loading capacity (>40%), low cytotoxicity and good biocompatibility [25].

MOF-5 (also known as IRMOF-1) is one of the most typical representatives of the MOFs family. It is a three-dimensional framework structure composed of terephthalic acid and metal cluster Zn_4_O (Figure 2) [26], firstly synthesized by Yaghi et al. [27]. MOF-5 has open skeleton structure, controlled pore structure and pore surface area and high thermal stability function [28,29,30], which has been widely studied in gas storage [31] and separation [32], electrochemistry [33], catalysis, and medicine [34]. In particular, the study of MOF-5 as drug carriers has received much attention in recent years. MOF-5 was firstly synthesized by solvothermal method, but the current research on MOF-5 as a drug carrier was mostly synthesized by the direct addition method. The difference between the materials synthesized by the two methods as drug carriers is not yet known. Yang et al. [35] used the direct addition method to prepare MOF-5, which had relatively high drug loading (capsaicin 0.592 g/g, 5-FU 0.315 g/g), and the release result was good. Liu et al. [36] used the direct synthesis method to prepare MOF-5, and the adsorption rate of curcumin arrived at 94.57%. The hollow structure of MOF-5 was synthesized by Lei Z et al. [37], the drug-loading rate of the modified nanoparticles was 42.43%, which achieved a more ideal sustained release effect. Tetracycline was encapsulated in the Carboxymethylcellulose/MOF-5/GO bio-nanocomposite, which showed effective protection against stomach pH and enhanced the long-time stability of drug dosing [38]. Therefore, as a new drug carrier, MOF-5 has a high drug loading and excellent release properties.

Herein, we synthesized MOF-5 by direct addition approach and solvothermal method, respectively. It was characterized by pharmaceutical methods and carried ORI via solvent adsorption method. The synthesis method which is more suitable for drug carrier was determined by comparison. The in vitro release properties were also examined. The flow chart of synthesis, drug-loading and in vitro release is shown in Figure 3. HepG2 cells were selected as cell models. The biosafety of MOF-5 and the antitumor activity after drug-loading were examined by MTT assay, 4′,6-diamidino-2-phenylindole (DAPI) staining and Annexin V/PI assay. Our results indicate that MOF-5, which synthesized by direct addition, has good biocompatibility. It has high drug loading and good sustained release effects as a drug carrier of ORI, and contributes to the anti-tumor effect of ORI. MOF-5 has significant potential as a drug carrier material.

## 2. Materials and Methods

### 2.1. Materials

Zinc nitrate hexahydrate (Zn(NO_3_)_2_·6H_2_O) was purchased from Shanghai Aladdin Biochemical Technology Co., Ltd. (Shanghai, China). Terephthalic acid (H_2_BDC) and dimethyl sulfoxide (DMSO) were obtained from Tianjin Guangfu Fine Chemical Research Institute (Tianjin, China). Triethylamine (TEA) was acquired from Tianjin Fuchen Chemical Reagent Factory (Tianjin, China). Trichloromethane (CHCl_3_) and *N*,*N*-Dimethylformamide (DMF) were got from Beijing Chemical Factory (Beijing, China). ORI was purchased from Nantong Feiyu Biotechnology Co., Ltd. (Nantong, China). Methanol (Chromatographic reagent grade) was obtained from Thermo Fisher Scientific (Shanghai, China). PBS and penicillin-streptomycin mixture were purchased from Solarbio (Beijing, China). High-glucose Dulbecco’s Modified Eagle Medium (DMEM) was obtained from Corning (Manassas, VA, USA). Fetal bovine serum was available from Zhejiang Tianhang Biotechnology Co., Ltd. (Zhejiang, China). 3-(4,5-dimethylthiazol-2-yl)-2,5-dipheny-ltetrazolium bromide (MTT) was obtained from Beijing Biodee Biotechnology Co., Ltd. (Beijing, China). 4′,6-diamidino-2-phenylindole (DAPI) and Annexin V-FITC Apoptosis Detection Kit were purchased from Shanghai Beyotime Biotechnology Co., Ltd. (Shanghai, China). All chemicals were of analytical grade or higher.

### 2.2. Characterization

#### 2.2.1. Scanning Electron Microscopy (SEM)

The surface morphology of the synthesized samples was identified using a Hitachi S-4700 cold field emission scanning electron microscope (Tokyo, Japan). High magnification SEM images of MOF-5-S and MOF-5-D can be found in the Appendix A. 

#### 2.2.2. X-Ray Diffraction (XRD)

X-ray diffraction measurements were investigated through a Rigaku UItima IV X-ray diffractometer (Tokyo, Japan). The measurements were performed using Cu-Kα (λ = 1.541nm) radiation at 40 kV and 40 mA in the scan range of 2θ from 5 to 50°.

#### 2.2.3. Thermogravimetric Analysis (TG)

Thermal decomposition was performed using a Mettler Toledo STARe system TGA/DSC3+ thermogravimetric analyzer (Zurich, Switzerland). The samples were heated in alumina pans from 30 °C to 600 °C with a heating rate of 10 °C/min under N_2_ flow.

#### 2.2.4. Fourier Transform Infrared Spectrometer (FTIR)

The functional group modifications of the particles were detected by a Thermo Fisher Nicolet-6700 Fourier transform infrared spectrometer (Waltham, MA, USA) at the wavelength range of 400–4000 cm^−1^.

#### 2.2.5. Dynamic Light Scattering (DLS)

The particle size distribution of the samples was measured through dynamic light scattering by a Malvern Nano Zetasizer S90 (Malvern, UK). The particle size distribution was evaluated using pure water as a dispersion medium at room temperature.

#### 2.2.6. Brunauer–Emmett–Teller (BET)

The specific surface area and pore size distribution of samples were measured by nitrogen adsorption/desorption at 77 K using a Belsorp-max BET Sorptometer (Bel Japan Inc., Tokyo, Japan).

### 2.3. Preparation of MOF-5

Direct addition approach: MOF-5 was prepared and activated according to previous reports [39,40]. Zn(NO_3_)_2_·6H_2_O (0.62 g) and H_2_BDC (0.16 g) were fully dissolved in 20 mL of DMF. And then, 11 mL of TEA was very slowly injected while stirring. The reaction mixture was sealed and stirred at room temperature for 2 h. After the reaction was done, the solid precipitates were collected by centrifugation, and then washed with DMF and CHCl_3_ for three times, respectively, and immersed into CHCl_3_ for 3 days. The solvent was changed once a day. Thereafter, the CHCl_3_ solvent was decanted and the solids dried at 120 °C for 12 h under vacuum. The obtained material was kept in a dryer and named MOF-5-D.

Solvothermal synthesis [41]: The amount of ligand and solvent was unchanged. The reactants were heated in a Shanghai Yushen Teflon-lined stainless-steel reaction vessel (Shanghai, China) at 135 °C for 24 h, and naturally cooled to room temperature. The post-treatment method was the same as above. The resulting material was named MOF-5-S.

### 2.4. Drug-Loading and In Vitro Release Studies

Solvent adsorption method was used to encapsulate ORI. ORI (90 mg) was dispersed in 6 mL methanol under ultrasonication. Then, the as-prepared MOF-5 nanoparticles (30 mg) were immersed in the drug solution and were stirred on a magnetic stirrer with 150 rpm at room temperature for 72 h. The ORI@MOF-5 was centrifuged and washed three times with methanol to get rid of excess ORI. The drug concentration in the supernatant was evaluated by a Thermo Scientific UltiMate 3000 HPLC (Waltham, MA, USA). ORI@MOF-5 was dried in the vacuum oven for removing residual methanol. ORI-loading capacity was calculated by using the following equation:(1)$$\mathrm{Loading}\;\mathrm{capacity}=\frac{M_{O}-M_{uO}}{M_{O}-M_{uO}+M_{M}}\times 100\%.$$

In Equation (1), M_O_, M_uO_, and M_M_ are the total amount of ORI, the un-loaded amount of ORI in the supernatant, and the amount of MOF-5, respectively.

In vitro release profiles of ORI@MOF-5 were obtained by the dialysis method. Typically, ORI@MOF-5 was put into dialysis bags and immersed in a certain volume of phosphate buffered saline (PBS) at pH 7.4, 5.5 and 2.0. And stirring with 100 rpm at 37 °C in a SHA-BA Constant temperature bath oscillator (Changzhou, China). Then at certain time intervals, adequate amounts of aqueous solution containing ORI were picked up and replaced with the same amount of fresh buffer. The amount of released drug was measured using HPLC. The corrected concentration and the percent of ORI released were expressed as:(2)$$C_{c}=C_{t}+\frac{v}{V}\overset{t-1}{\underset{0}{\sum }}C_{t},$$
(3)$$\mathrm{Drug}\;\mathrm{release}=\frac{M_{R}}{M_{L}}\times 100\%.$$

In Equation (2), C_c_ stands for the adjusted concentration of ORI at time t, C_t_ is the measured concentration at t, v is the volume of the derived samples, and V is the volume of release solution. M_R_ and M_L_ are the amount of released drug and loaded drug (equation 3), respectively.

### 2.5. Cell Culture

HepG2 cells were obtained from Guangzhou Jeniobio Biotechnology (Beijing, China). The cells were cultured in a high-glucose DMEM medium containing with 10% FBS and antibiotics (100 µg/mL penicillin and 100 µg/mL streptomycin) for 2–3 days at 37 °C under a humid atmosphere with 5% CO2. Before passaged, they were harvested by trypsin and resuspended in fresh medium.

### 2.6. Cell Viability

The in vitro cytotoxicity of MOF-5, free ORI and ORI@MOF-5 towards HepG2 cells was assessed by the MTT assay. The 96-well plates were used to culture HepG2 cells at a density of 5000 cells/well in 100μL overnight. Then, cells were treated with different concentration of MOF-5, free ORI and ORI@MOF-5 for 24h. Following, the MTT solutions (5 mg/mL, 20μL) were added into each well and further incubated at 37 °C for 4 h. After that, the medium was discarded and DMSO solvent (200 μL/well) was added to solubilize the formazan crystals. The absorbance at a wavelength of 490 nm were measured with a BMG SPECTROstar Nano High-throughput microplate UV spectrophotometer (Ortenberg, Germany) after shaken for 10 min.

### 2.7. Nuclear Morphology

HepG2 cells were cultured at a density of 4 × 10^5^ cells/well in six-well plates and incubated overnight. After treatment with different concentration of MOF-5, free ORI and ORI@MOF-5 for 24 h. The culture supernatant was removed and the cells were fixed with 4% paraformaldehyde (500 μL) for 10 min at room temperature and then washed twice with PBS. Then cells were incubated with DAPI solution (10 µg/mL, 0.8 mL) in the dark for 10min and washed twice with PBS again. Morphological changes of nuclei fragmentation and chromatin condensation were photographed using a Nikon Research Inverted Microscope ECLIPSE Ts2R (Tokyo, Japan).

### 2.8. Flow Cytometric for Apoptosis Analysis

The Annexin V-FITC/PI apoptosis detection kit was used to detect the apoptosis of HepG2 cells by flow cytometry according to the manufacturer’s instructions. HepG2 cells were seeded (4 × 10^5^ cells/well) into 6-well plates and incubated overnight. And then incubated with various concentrations of MOF-5 and ORI@MOF-5 for 24 h. Following, the cells were harvested and washed with PBS. Then, the cells were suspended in 195 µL Annexin V binding buffer, stained with 5 µL FITC-labelled Annexin V and 10 µL PI, hatching 20 min in the dark, and then immediately analyzed by a BD FACSCanto II Flow cytometry (Franklin Lakes, NJ, USA).

### 2.9. Statistical Analysis

The statistical analysis was carried out with further statistical methods of single factor analysis of variance, multivariate comparison, and a non-parametric test by using SAS 9.4 software (Beijing, China). Each assay was carried out in triplicate in a parallel manner, and the results were expressed as mean ± SD. P values < 0.05 were regarded as significant.

## 3. Results

### 3.1. Synthesis and Characterization

MOF-5 was synthesized according to the previously described two distinct methods. Before pharmaceutical investigation and biological assays, several techniques were used to characterize the materials, including SEM, XRD, FTIR, TG, and DLS.

The surface morphology of MOF-5-S and MOF-5-D was revealed by SEM and showed in Figure 4A,B. MOF-5-S has a regular hexahedral structure with smooth surface. The particle size distribution is on the order of micrometers, and some of them have angle deficiency. In contrast, MOF-5-D is a nano-sized irregular flaky particle. This may be related to their growth rate. MOF-5-S was slowly synthesized in a high temperature and pressure environment, while MOF-5-D was rapidly grown within a few hours by adding TEA. Therefore, they have obvious differences in appearance and size. SEM images show that prepared MOF-5-D tends to aggregate due to the small size effect caused by water molecules on metal clusters in particles.

The XRD pattern of the particles has been compared with simulated MOF-5 XRD data as presented in Figure 4C. Acceptable matches are observed for the diffraction peaks of the two sets of samples, which indicates that both methods could synthesize well-structured materials. It has been reported that the intensity of the diffraction peak at 6.8° is directly proportional to the amount of pore filling material such as solvent [42]. The 6.8° peak intensity of MOF-5-D is weak in Figure 4C, which showed that most of the solvents in the pore were removed in the post-treatment process. Moreover, diffraction peaks appeared at 8.8°, indicating that the sample may absorb some water, resulting in slight structural changes.

MOF-5-S and MOF-5-D were characterized by FTIR and TG to determine their functional groups, chemical structure and thermal stability. As shown in Figure 4D, the peak shapes and positions in the two spectra are similar, indicating that they have similar functional groups and chemical structures. The two peaks around 800 cm^−1^ are caused by C–H vibration on the benzene ring, which proves that the benzene ring is 1,4-substituted. The strong characteristic peaks at 1590 cm^−1^ and 1486 cm^−1^ are the stretching vibration peaks of –COO bond. The obvious wide absorption peak near 3400 cm^−1^ corresponds to the absorption peak of water molecules in the sample. The sharp peak at 3610 cm^−1^ is the combination of metal center with water molecule, which confirms the results of the XRD pattern. The FTIR spectra are consistent with those reported in previous literature [43], indicating that both methods are feasible.

TG analysis (Figure 4E) shows that the thermal stability of the two samples is good and the skeleton is stable before 400 °C. At 400–500 °C, the organic skeleton is decomposed into water and carbon dioxide. A small amount of weight loss before 350 °C is the detachment of solvent and water molecules inside or outside the pore size of the samples. Finally, the weight is no longer reduced and the residue is ZnO.

The particle size distribution of MOF-5-S and MOF-5-D is shown in Figure 5. As shown in Figure 5A, 87% of MOF-5-S particles are distributed at 1794 nm, and the remaining small particles may be debris particles. The particle size of MOF-5-D is 260 nm and it has uniform particle size distribution, as indicated in Figure 5B. N_2_ adsorption−desorption experiment of MOF-5-S and MOF-5-D was carried out and shown in Figure 5C, and the results are in good agreement with a nano porous materials’ structure. The Brunauer–Emmett–Teller (BET) specific surface area of MOF-5-S is 580.35 m^2^/g and the pore volume is 0.2918 cc/g. Those of MOF-5-D are 736.92 m^2^/g and 0.9917 cc/g, respectively. According to BET test results, the specific surface area and pore volume of MOF-5-D are significantly higher than those of MOF-5-S. This may be due to their different synthesis methods. The particle size distribution and BET results showed that MOF-5-D synthesized by direct addition method was more suitable as drug carrier.

Through single factor experiments, ORI was loaded with MOF-5-S and MOF-5-D, respectively, under the same conditions. The loading capacity was compared and the optimum synthesis method was selected. The drug loading of MOF-5-S is 14.23% ± 1.12%, while that of MOF-5-D is as high as 31.36% ± 2.36%. Based on the above experimental results, both methods can successfully synthesize MOF-5. However, the particles synthesized by direct addition method are more suitable as carrier materials for ORI, which will be used for subsequent experiments.

### 3.2. Drug Loading

High drug-loading of drug carriers plays an important role in increasing therapeutic efficiency. Using L_9_ (3^4^) orthogonal experimental design, and three factors affecting drug loading were selected: the ratio of MOF-5 to ORI, drug-loading time and ORI concentration. The drug loading was 52.58% ± 0.59% (Equivalent to 1.11 g (3.04mmol) of ORI per gram of MOF-5) under the optimum conditions: MOF-5: ORI (1:3), 4 days, 15 mg/mL. ORI@MOF-5 was characterized by FTIR and TG, as illustrated in Figure 6.

It can be obviously seen from the FTIR comparison chart that ORI@MOF-5 contains not only the characteristic peaks of MOF-5, but also peaks of ORI, such as peaks at 3000–2800 cm^−1^, 1700 cm^−1^, and 1200–900 cm^−1^. The result shows that ORI@MOF-5 successfully encapsulates ORI while preserving the structure of MOF-5 itself.

The results of TG analysis show that ORI begins to decompose at 250 °C and decomposes completely at 600 °C. MOF-5 mainly loses weight between 400–500 °C. After 500 °C, the weight does not lose any more. ORI@MOF-5 begins to lose weight at 250 °C and the weight is no longer reduced at 500 °C, which proves that ORI is successfully loaded.

The results show that ORI@MOF-5 has good crystallinity. The strong peak at 8.8° is assumed to be the result of the combination of carbonyl groups and metal centers in ORI. Whether ORI enters the MOF-5 aperture requires BET experiments to reveal.

The BET specific surface area of ORI@MOF-5 is 48.189 m^2^/g and the pore volume is 0.2175 cc/g. They are significantly lower than the blank material, indicating the successful loading of ORI.

### 3.3. In Vitro Release

The in vitro release properties of ORI@MOF-5 were investigated under different pH conditions, as shown in Figure 7A. Before 36h, the release rate of ORI@MOF-5 is slightly different under three different pH conditions, with pH 5.5 being the fastest and pH 2.0 being slower. After reaching 36h, the release rate and total amount of different pH values tend to be the same. The cumulative release has exceeded 50% at 12 h. The maximum cumulative release was 87.18% ± 1.26% at pH 7.3, 87.02% ± 1.28% at pH 5.5 and 87.89% ± 0.71% at pH 2.0. It is indicated that the in vitro release behavior of ORI@MOF-5 is less affected by pH.

Curve fitting was performed on in vitro release results using zero order equation, first order equation, Higuchi equation and Ritger-Peppas equation, as indicated in Figure 7B. In vitro release kinetic study is generally fitted by zero order and first order equations. The release mechanism is usually fitted using the Higuchi equation. However, for the dissolution type skeleton system such as MOF material, drugs diffuse from the skeleton material, while the skeleton is also in the dissolution, which accelerates the release of the drug. In this case, there is a substantial deviation in fitting only by Higuchi equation. Peppas et al. proposed an empirical equation that describes the drug release law of the degradable system, named Ritger–Peppas equation. Therefore, we used the zero-order, first-order, Higuchi and Ritger–Peppas equations to fit the release curve for the initial 36 h. Under three pH conditions, ORI@MOF-5 fits best with the First-order release equation, indicating that the drug-loaded particles have good sustained release effect. For the release mechanism, Ritger–Peppas model fits well. And the diffusion index fitting results of the Ritger–Peppas model, ie, the lnt coefficient, are all between 0.45 and 0.89, indicating that the release behavior of the particles is consistent with non-Fick’s diffusion. ORI@MOF-5 in vitro drug release is the joint result of molecular diffusion and skeleton erosion.

### 3.4. Biocompatibility of Nanocarrier

Before comparing the cytotoxic effects of free ORI and ORI@MOF-5, the biocompatibility of blank nanocarrier should be comprehensively evaluated. Firstly, the classical MTT assay was carried out on the HepG2 cells. According to Figure 8A, the cells reveal high viability after 24 h incubation of different concentration dosages MOF-5 (5, 10, 15, 20, and 25 μg/mL), indicating that MOF-5 is non-toxic at the experimental dose.

Then, DAPI staining was used to determine whether MOF-5 had an effect on the nuclear morphology of HepG2 cells. After 24 h treatments of HepG2 cells with different concentrations of MOF-5, the nucleus morphological of each group are in good shape (Figure 8B), without rupture, deformation, or other phenomena, which confirms the good biocompatibility of MOF-5.

To further verify the cytotoxicity, the Annexin V/PI assay was used to detect the effect of MOF-5 on apoptosis. The results are shown in Figure 9. After being treated with different doses of MOF-5 for 24 h, there is no significant change in the percentage of viable cells, early apoptotic cells, and dead cells. There is no significant induction of apoptosis at the experimental dose, indicating that MOF-5 is biocompatible.

### 3.5. In-Vitro Toxicity Study of Ori@MOF-5

HepG2 cells were treated with different doses of free ORI (5, 10, 15, 20, and 25 μg/mL) and ORI@MOF-5 (Drug loading 52.86%) containing the same dose of ORI for 24 h, respectively. The results of toxicity comparison by MTT assay are shown in Figure 10A. Cell viability decreases with the increase of drug concentration, indicating that both ORI and ORI@MOF-5 show significant cytotoxicity in HepG2 cells in a dose-dependent manner. At the equivalent ORI concentration, the toxicity of ORI @MOF-5 is slightly lower than that of free ORI. And the IC_50_ of them are ORI: 11.08 ug/mL and Ori@MOF-5: 22.99 ug/mL, respectively. Analogous results have been reported in previous studies [44,45]. Due to the interactions between the drug and carrier, ORI is slowly and continuously released from ORI@MOF-5, resulting in reduced toxicity. Although ORI@MOF-5 has a weaker killing effect on HepG2 cells than free ORI, its sustained release reduces the toxicity and side effects of drugs, which is more conducive to clinical use.

After treatment with ORI@MOF-5 at a concentration of 22.99 ug/mL (IC50) for 24 h, HepG2 cells were stained with DAPI. As shown in Figure 10B, compared with the control group, the nuclei of the experimental group occur significantly fragmentation and morphological change, with the common morphological characteristics of apoptosis.

Moreover, the effect of ORI@MOF-5 on apoptosis of HepG2 cells was investigated by Annexin V/PI double staining assay. As illustrated in Figure 11, after treatment with different concentrations of ORI@MOF-5 (containing ORI 0-25 μg/mL) for 24h, the apoptosis of HepG2 cells increases with the increase of concentration, which is consistent with the MTT results. Apoptotic cells account for 94.97% at the highest dose, indicating that ORI@MOF-5 has good anti-tumor effect.

## 4. Discussion

In this study, MOF-5 was selected as the carrier of the antitumor active ingredient ORI for the first time. ORI@MOF-5, a nano-scale sustained release drug delivery system, has been successfully prepared. It has superior biocompatibility, high drug loading, good sustained release effect and excellent anti-tumor effect.

Here, nano-MOF-5 was successfully synthesized by direct addition method, which has a stable structure, good crystal shape and good thermal stability. These are consistent with the results of Huang L, et al. [39]. Compared with the micron-sized MOF-5 synthesized by solvothermal method, we found that MOF-5 synthesized by direct addition method had better drug-loading effect on ORI. MOF-5-D has small size, high specific surface area and pore volume, which promote drug inclusion. At present, no scholars have conducted research in this area.

By comparing the results of IR, TG, XRD, and BET analysis before and after drug loading, it was found that ORI was successfully loaded into MOF-5. The results of IR showed that the drug-loaded materials contained common characteristic peaks of blank materials and free drugs. XRD results show that the crystallinity of ORI@MOF-5 is good, and the strong peak at 8.8° is the combination of carbonyl group in ORI and the metal center in MOF-5. At the same time, the specific surface area and porosity of the material decreased significantly after loading, which indicated the loading of the drug.

Then, drug-loading and in vitro release were performed. The drug loading of ORI@MOF-5 is as high as 52.86% ± 0.59%, which is much higher than other types of drug delivery systems, such as gold nanoparticles [46] or graphene oxides [47]. In our previous study [48], ORI was loaded by MIL-53 (Fe), and the drug loading was less than 50% under the same calculation method. Clinically, high drug loading can improve the efficiency of treatment and reduce the dosage of drug, which is of great significance.

The results showed that ORI@MOF-5 has a nice sustained release effect under different pH conditions. Its release is basically not affected by pH, and the cumulative release is 87% in 60 h. This characteristic is linked to the absence of exposed groups in the carrier material itself, which indicates that the drug delivery system can adapt to a variety of pH environments and is easier to design in clinical use.

Good biocompatibility is the foundation of the application of carrier materials. In our previous study [49], MTT, DAPI staining, and an apoptosis experiment were used to detect the in vitro toxicity of MIL-100 (Fe), which could reflect the toxicity of the drug comprehensively. Here, the safety evaluation results show that the safe dose of MOF-5 reaches 25 μg/mL, which has little effect on cell viability, nuclear morphology, and cell apoptosis under safe doses. Subsequent experiments were conducted under safe doses.

The efficacy of the drug-loading system determines the value of its clinical value. Compared to free ORI, ORI@MOF-5 has a slightly lower efficacy at the same dose. A similar situation has been reported in the literature [50]. Despite the fact that the drug effect is slightly lower, its sustained release property plays an important role in reducing toxicity and side effects and has more clinical value.

In this study, only the 24 h toxicity study was performed because after 24 h, the release rate was slow and the cumulative release was slightly below the maximum. What’s more, the maximum concentration of cell survival was observed to be less than 10% at 24h. Therefore, a longer investigation was not carried out. The characteristics of ORI@MOF-5 drug-loading system are studied only through in vitro experiments, which is not a substitute for in vivo experiments. However, as a conventional evaluation method, it can be used as a basis for future in vivo experiments. In the next step, more different cell lines will be used and animal experiments will be carried out to study the safety, release characteristics, antineoplastic efficacy and other research in vivo, so as to provide better reference for clinical applications.

## 5. Conclusions

In the current study, nano-MOF-5 was synthesized and loaded with the anti-tumor active ingredient ORI. The results show that nano-MOF-5 has good biocompatibility and biodegradability. Drug-loading and in vitro release experiments confirm that ORI@MOF-5 has high drug loading and good sustained release properties. MTT assay, DAPI staining and apoptosis experiments suggest that ORI@MOF-5 has significant cytotoxicity and apoptosis effect on HepG2 cells. As a result, ORI@MOF-5 has the potential to become a new anti-cancer sustained release preparation.

## Figures and Tables

**Figure 1 molecules-24-03369-f001:**
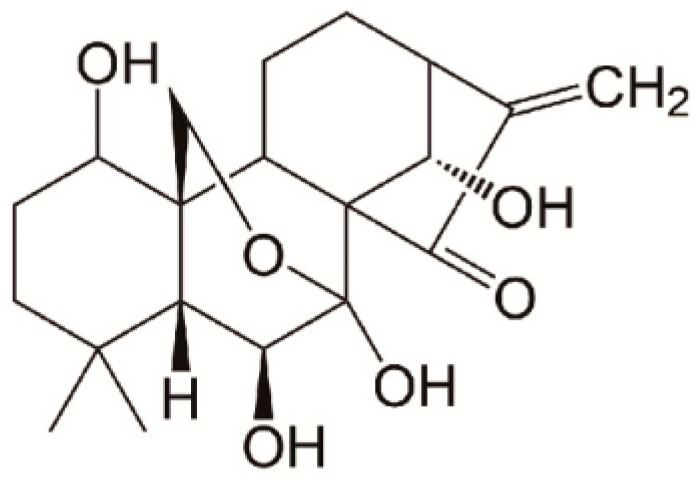
Chemical structure of oridonin.

**Figure 2 molecules-24-03369-f002:**
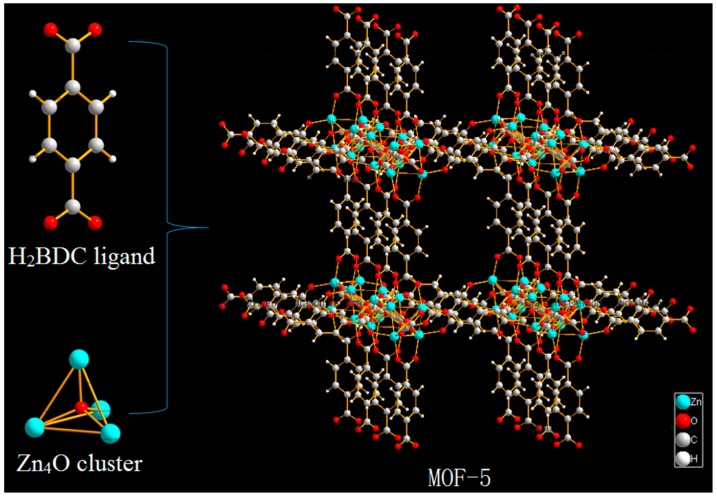
Schematic illustration of the construction of metal organic framework material (MOF-5).

**Figure 3 molecules-24-03369-f003:**
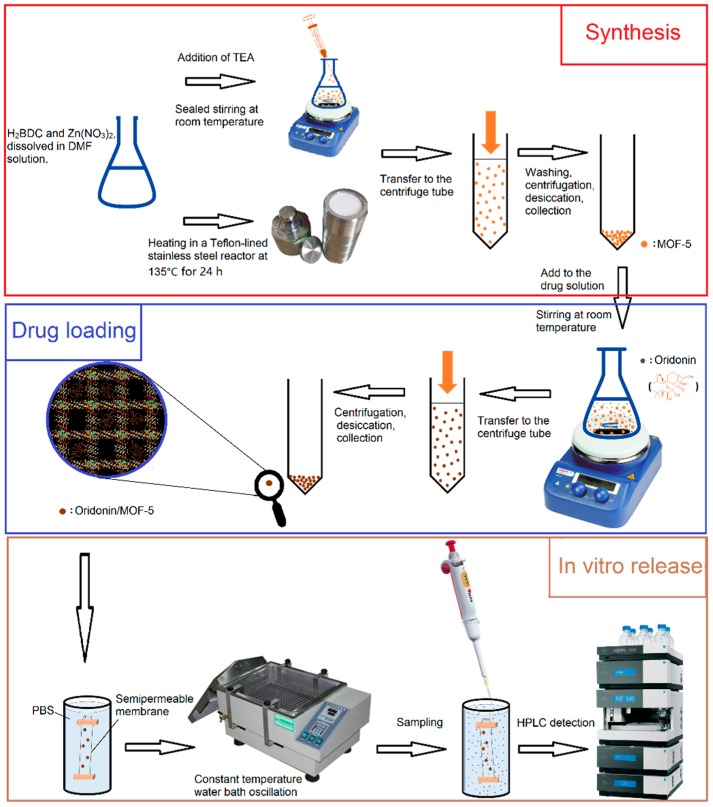
The flow chart for synthesis, drug-loading, and in vitro release.

**Figure 4 molecules-24-03369-f004:**
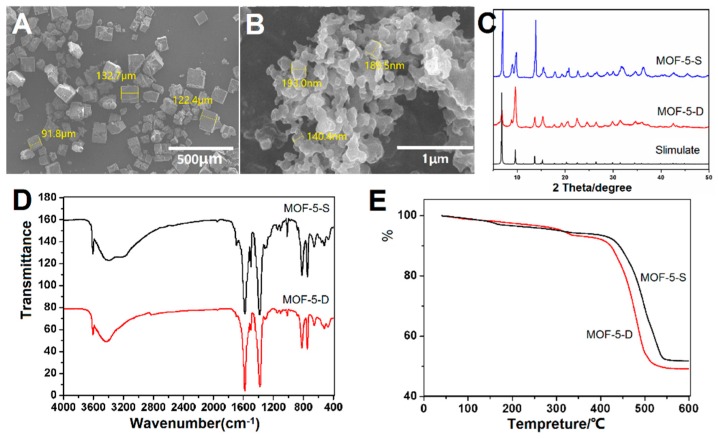
Characterization of MOF-5 synthesized by two methods: (**A**) Scanning Electron Microscopy (SEM) Image of MOF-5-S; (**B**) SEM Image of MOF-5-D; (**C**) X-ray diffraction (XRD) patterns of MOF-5-S and MOF-5-D (**D**) Fourier Transform Infrared Spectrometer (FTIR) spectra of MOF-5-S and MOF-5-D; (**E**) Thermogravimetric (TG) analysis of MOF-5-S and MOF-5-D.

**Figure 5 molecules-24-03369-f005:**
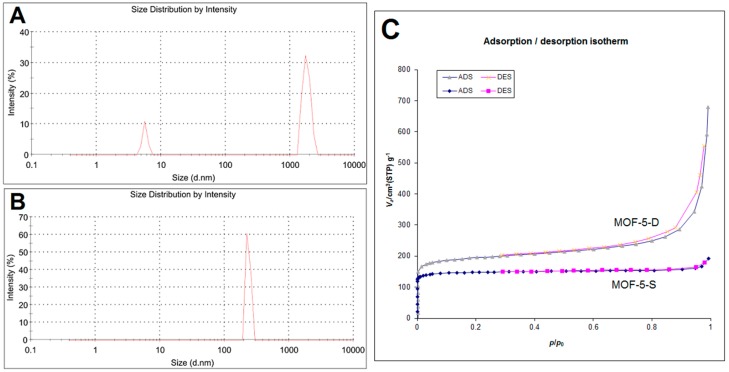
Particle size distribution of MOF-5-S (**A**) and MOF-5-D (**B**); (**C**) Brunauer–Emmett–Teller (BET) isotherm of MOF-5-S and MOF-5-D at 77 K.

**Figure 6 molecules-24-03369-f006:**
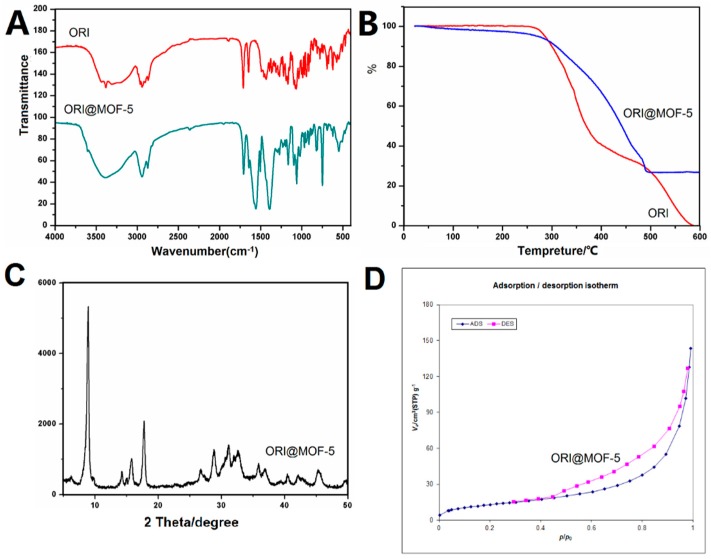
(**A**) FTIR spectra of ORI and ORI@MOF-5; (**B**) TG analysis of ORI and ORI@MOF-5; (**C**) XRD patterns of ORI@MOF-5; (**D**) BET isotherm of ORI@MOF-5.

**Figure 7 molecules-24-03369-f007:**
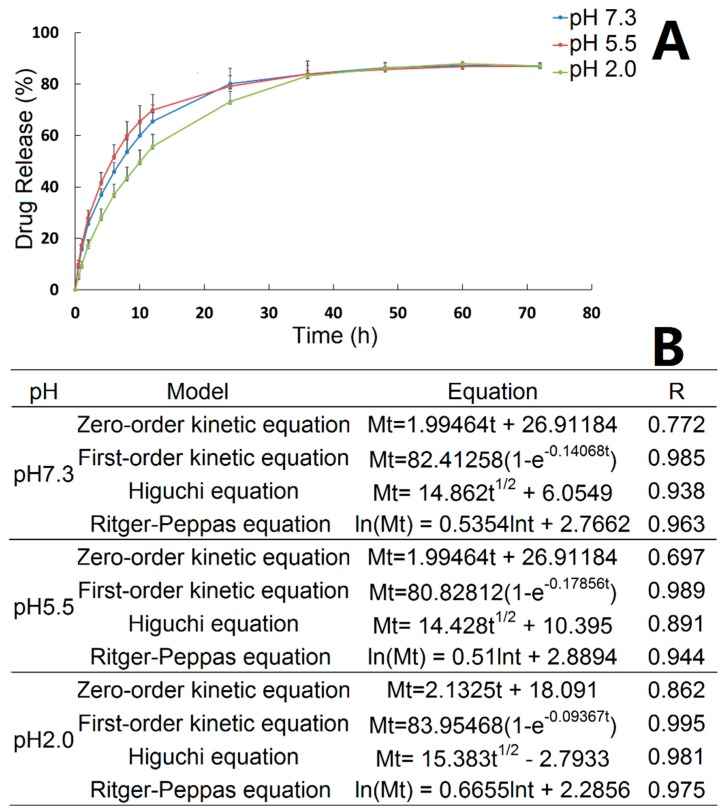
(**A**) The cumulative release of ORI from ORI@MOF-5 at three pH values; (**B**) Fitting results of equation.

**Figure 8 molecules-24-03369-f008:**
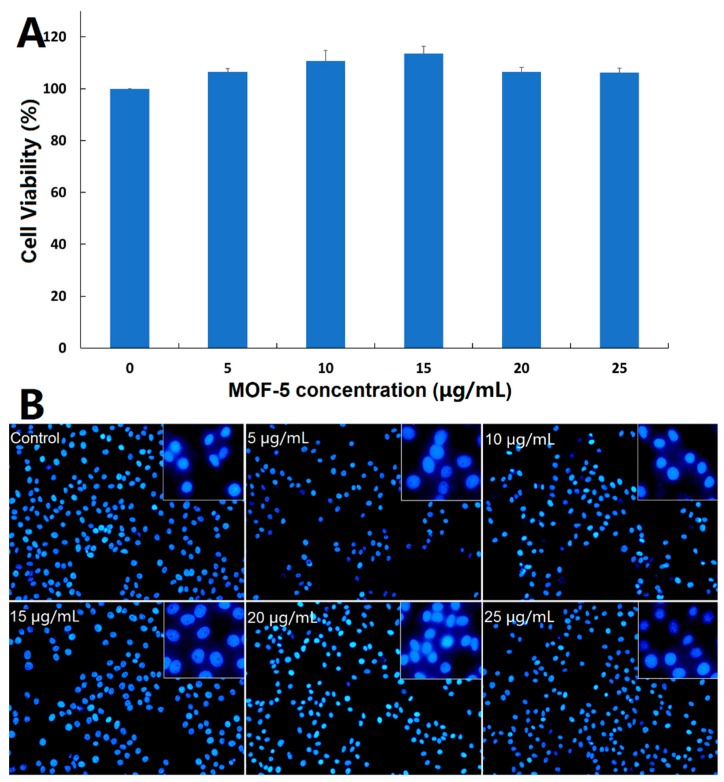
(**A**) Cytotoxicity study of MOF-5 on HepG2 cells determined by MTT assay; (**B**) Fluorescent microscopic images of DAPI-stained HepG2 cells following 24-h treatment with different concentrations of MOF-5.

**Figure 9 molecules-24-03369-f009:**
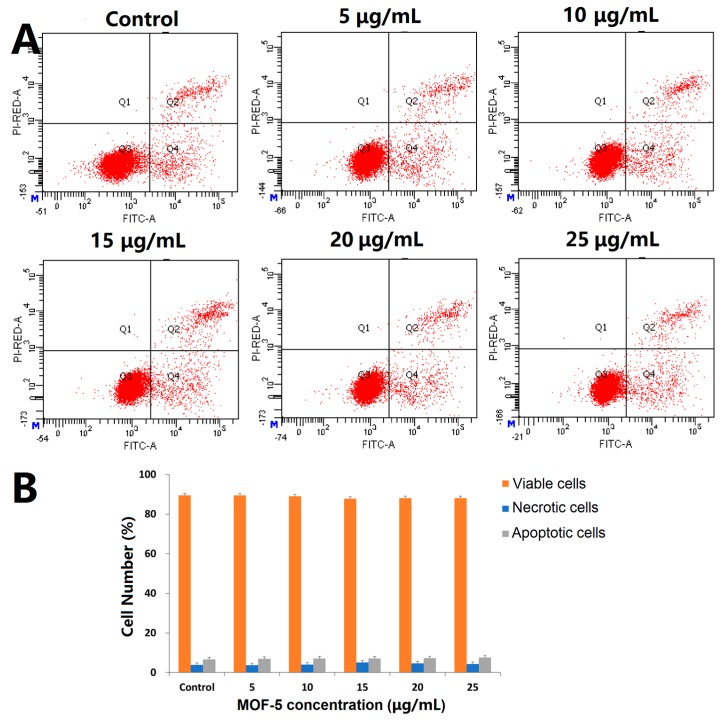
(**A**) Apoptosis assays for HepG2 cells after treatment with different concentrations of MOF-5 for 24 h; (**B**) Statistical analysis of cell apoptosis.

**Figure 10 molecules-24-03369-f010:**
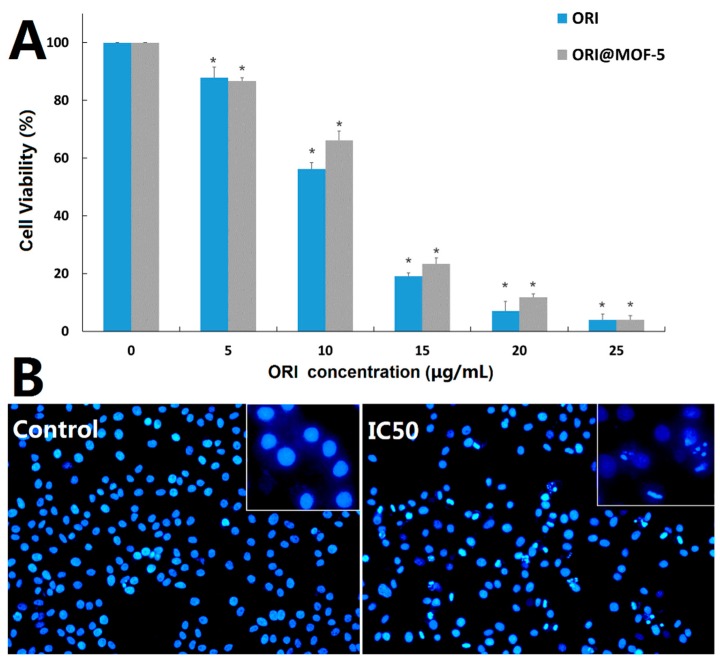
(**A**) In vitro cell viabilities of HepG2 cells after being incubated for 24 h with various concentrations of ORI and ORI@MOF-5; (**B**) Fluorescence microscopic images of HepG2 cells stained with DAPI after 24-h treatment with ORI@MOF-5 (at IC50 value).

**Figure 11 molecules-24-03369-f011:**
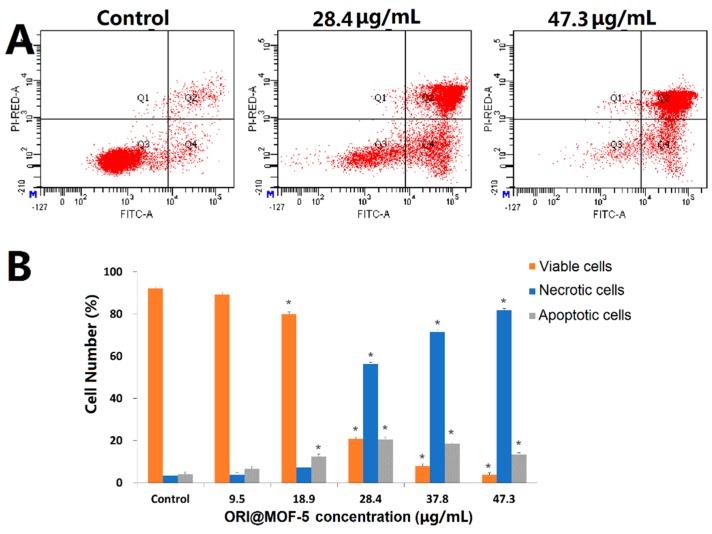
(**A**) Apoptosis assays for HepG2 cells after treatment with different concentrations of ORI@MOF-5 for 24 h; (**B**) Statistical analysis of viable, necrotic, and apoptotic HepG2 cells.

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
