# Peer review of "Investigation of Metal-Organic Framework-5 (MOF-5) as an Antitumor Drug Oridonin Sustained Release Carrier"

_molecules, 2019, doi:10.3390/molecules24183369_

Round 1

Reviewer 1 Report

The results presented in the manuscript could be interesting and useful for readers. However some improvements should be done before publication.

Since both MOF-5 synthesis methods (direct addition approach and solvothermal method) used by authors in this study are not original and developed previously, I highly recommend to change the title of the manuscript to “Investigation of Metal-Organic Framework-5 (MOF-5) as Antitumor Drug Oridonin Sustained release Carrier”. Introduction. The recent publications should be cited and discussed in the text: (1) D.D. Chai et al. Delivery of Oridonin and Methotrexate via PEGylated Graphene Oxide. ACS APPLIED MATERIALS & INTERFACES. Volume: 11, Issue: 26 (2019) Pages: 22915-22924 and (2) Z. Karimzadeh et al. Carboxymethylcellulose/MOF-5/Graphene oxide bio-nanocomposite as antibacterial drug nanocarrier agent. BIOIMPACTS. Volume: 9, Issue: 1 (2019) Pages: 5-13. Caption for Figure 1 should be changed slightly to “Figure 1. Chemical structure of oridonin”. Figure 4, SEM micrographs. For precise comparison the both SEM images should be presented at the same magnification. Moreover, the SEM images should be presented at higher magnifications, since authors claim about possible formation of nanosized particles. Figure 6. The FTIR spectrum and TG curve of MOF-5 should be removed from appropriate figures, since they are already presented in Figure 4. In conclusion, the manuscript could be accepted for publication after major revision.

Author Response

Point 1: Since both MOF-5 synthesis methods (direct addition approach and solvothermal method) used by authors in this study are not original and developed previously, I highly recommend to change the title of the manuscript to “Investigation of Metal-Organic Framework-5 (MOF-5) as Antitumor Drug Oridonin Sustained release Carrier”.

Response 1: Dear expert reviewer, thank you for your valuable comments. We have changed the title of the manuscript to “Investigation of Metal-Organic Framework-5 (MOF-5) as Antitumor Drug Oridonin Sustained release Carrier”.

Point 2: Introduction. The recent publications should be cited and discussed in the text: (1) D.D. Chai et al. Delivery of Oridonin and Methotrexate via PEGylated Graphene Oxide. ACS APPLIED MATERIALS & INTERFACES. Volume: 11, Issue: 26 (2019) Pages: 22915-22924 and (2) Z. Karimzadeh et al. Carboxymethylcellulose/MOF-5/Graphene oxide bio-nanocomposite as antibacterial drug nanocarrier agent. BIOIMPACTS. Volume: 9, Issue: 1 (2019) Pages: 5-13. Caption for Figure 1 should be changed slightly to “Figure 1. Chemical structure of oridonin”.

Response 2: Dear expert reviewer, we have taken your advice and the two recent papers were cited and discussed in the introduction section. Caption for Figure 1 has been corrected.

Point 3: Figure 4, SEM micrographs. For precise comparison the both SEM images should be presented at the same magnification. Moreover, the SEM images should be presented at higher magnifications, since authors claim about possible formation of nanosized particles.

Response 3: Dear expert reviewer, your advice is very important. However, in this paper, the size of the two groups of materials is quite different (MOF-5-S is micron, MOF-5-D is nanometer). If both of them are amplified to the same large magnification, it is hardly to ensure that the complete morphology of the materials can be observed. Therefore, in order to show the appearance of the material intuitively, we choose different magnification according to the actual situation, and mark it in the figure. We will upload higher magnification pictures in supplementary documents for reference.

Point 4: Figure 6. The FTIR spectrum and TG curve of MOF-5 should be removed from appropriate figures, since they are already presented in Figure 4.

Response 4: Dear expert reviewer, the FTIR spectrum and TG curve of MOF-5 have been removed from appropriate figures.

Finally, I would like to sincerely thank you for your careful review and valuable suggestions.

Reviewer 2 Report

Ni and co-workers’ paper entitled „Synthesis and Characterization of Metal-Organic  Framework-5 (MOF-5) and Study as Antitumor Drug  Oridonin Sustained release Carriers" reported that synthesis of nano-MOF-5 and incorporation of active diterpenoid inside of MOFs.

The data are interesting, but the content was not a significant novelty. The procedure and techniques involved in the experiment and analysis are rather routine. The manuscript is not suitable for publication in its present form. The following comments can be considered by the authors for improving its manuscript.

First of all, the synthesis of the material in DMF is of great concern for the use of this material as bio-application! Authors should determine the exact size of MOF crystallites on Fig 4a and b. In Figure 4C, authors showed the PXRD of simulated MOF 5 why is different for MOF-5-D and MOF-5-S? What is the maximum possible loading of ORI? Authors should conduct an experiment of Solid–liquid adsorption isotherm (see Chem. Commun.,2011,47, 11751) On the basis of TG, the authors should propose a form of the composite. What is the stability of the material after 72 hours? Specific experiments such as IR and XPRD should be presented. What about the stability of MOF-5 in the pH range of 2-7.3? Such an experiment should also be tested.

Author Response

Point 1: First of all, the synthesis of the material in DMF is of great concern for the use of this material as bio-application! Authors should determine the exact size of MOF crystallites on Fig 4a and b.

Response 1: Dear expert reviewer, thank you for your comments. The particle size distribution of the samples was measured through dynamic light scattering and showed in figure 5. The particle size of MOF-5-D is 260 nm and the particle size of MOF-5-S is about 1800 nm.

Point 2: In Figure 4C, authors showed the PXRD of simulated MOF 5 why is different for MOF-5-D and MOF-5-S?

Response 2: In this paper, the materials were synthesized and activated at high temperature. It has been reported that the intensity of the diffraction peak at 6.8° is directly proportional to the amount of pore filling material such as solvent.( Burgaz E, Erciyes A et al. Synthesis and characterization of nano-sized metal organic framework-5 (MOF-5) by using consecutive combination of ultrasound and microwave irradiation methods[J]. Inorganica Chimica Acta, 2019, 485:118-124.) The 6.8° peak intensity here is weak and it is almost invisible in figure 4C, which showed that most of the solvents in the pore were removed in the post-treatment process. Moreover, strong diffraction peaks appeared at 8.8°, indicating that the sample may absorb some water, resulting in slight structural changes.

Point 3: What is the maximum possible loading of ORI? Authors should conduct an experiment of Solid–liquid adsorption isotherm (see Chem. Commun.,2011,47, 11751)

Response 3: Dear expert reviewer, your opinion is very pertinent. However, considering the clinical application, the orthogonal experimental design was used to investigate the maximum load in this study. Using L9(34) orthogonal experimental design, and three factors affecting drug loading were selected: the ratio of MOF-5 to ORI, drug-loading time and ORI concentration. The drug loading was 52.58±0.59% under the optimum conditions: MOF-5: ORI (1:3), 4 days, 15 mg/mL, according to the calculation formula under 2.4 Drug-loading and in vitro release studies. Namely, a maximum loading value of 1.11 g (3.04mmol) of ORI per gram of MOF-5.

Point 4: What is the stability of the material after 72 hours? Specific experiments such as IR and XPRD should be presented.

Response 4: Dear expert reviewer, your question is crucial. The stability of materials has always been the focus of scholars’attention. MOF-5 is a kind of MOFs material which has been well studied. It has good stability in dry environment. However, it has been proved that the coordination bonds of MOF-5 will be slowly destroyed by water molecules in humid environment, leading to the slow collapse of the skeleton.( Arjmandi M , Chenar M P et al. Influence of As-Formed Metal-Oxide in Non-Activated Water-Unstable Organometallic Framework Pores as Hydrolysis Delay Agent: Interplay Between Experiments and DFT Modeling[J]. Journal of Inorganic and Organometallic Polymers and Materials, 2019, 29:178-191.) Drug release is accompanied by collapse of material structure. The materials remained after 72 hours contain decomposed products and MOF-5 which have not yet been decomposed. 3.3. In vitro release results show that in vitro drug release is the joint result of molecular diffusion and skeleton erosion. Therefore, this paper does not do the corresponding inspection.

Point 5: What about the stability of MOF-5 in the pH range of 2-7.3? Such an experiment should also be tested. 

Response 5: Dear expert reviewer, thank you for your valuable comments. The in vitro release behavior of MOF-5 under three different pH conditions (pH=2.0; 5.5; 7.4) was investigated. The results showed that the release behavior of MOF-5 was almost unaffected by different pH conditions, indicating that the pH of MOF-5 was insensitive. Because of the poor water stability of MOF-5, the structural stability of MOF-5 under different pH conditions can not be determined by characterization, so its pH sensitivity can only be deduced from in vitro release. It can also be judged from the skeleton structure without active groups that pH has little effect on MOF-5.

From your comments, it’s obvious that you are an expert in the field of materials, and there is lots of shortage in our study which need your instruction. At last, I want to thank you sincerely for your suggestions and I feel so sorry that so much of your precious time was wasted on our paper revision.

Reviewer 3 Report

The authors showed that nano-MOF-5 prepared by direct addition method had complete structure, uniform size and good biocompatibility, and was suitable as ORI carrier. They also mentioned that ORI@MOF-5 has a slightly lower efficacy at the same dose compared to free ORI, but its sustained release property plays an important role in reducing toxicity and side effects and has more clinical value. I agree with their opinion that ORI@MOF-5 has the potential to become a new anti-cancer sustained release preparation. The article is well organized and easy to follow. I would like to recommend this article publish on Molecules after minor revision.

Revision points:

1. The authors did in vitro release studies of ORI@MOF-5 in PBS solution. I am wonder what about the stability of MOF-5 in PBS? They can provide PXRD of MOF-5 in PBS.

2. After all ORI are released from ORI@MOF-5, what about stability of the remaining MOF-5? Is the remaining MOF-5 decomposed after releasing? They can also provide PXRD of the remaining MOF-5 after in vitro release studies.

Author Response

Point 1: The authors did in vitro release studies of ORI@MOF-5 in PBS solution. I am wonder what about the stability of MOF-5 in PBS? They can provide PXRD of MOF-5 in PBS.

Response 1: Dear expert reviewer, your question is crucial. The water stability of some MOFs has long attracted the attention of scholars. In particular, MOF-5 has excellent properties, but it has been proved that the coordination bonds of MOF-5 will be slowly destroyed by water molecules in humid environment, leading to the slow collapse of the skeleton. (De?Toni M , Jonchiere R et al. How Can a Hydrophobic MOF be Water-Unstable? Insight into the Hydration Mechanism of IRMOFs[J]. Chemphyschem, 2012, aop.) Therefore, there is no such investigation in this paper, but 3.3. In vitro release results show that in vitro drug release is the joint result of molecular diffusion and skeleton erosion. It shows that the water instability of the material is beneficial to the drug release in practical application.

Point 2: After all ORI are released from ORI@MOF-5, what about stability of the remaining MOF-5? Is the remaining MOF-5 decomposed after releasing? They can also provide PXRD of the remaining MOF-5 after in vitro release studies.

Response 2: The materials remained after drug release contain decomposed products and MOF-5 which have not yet been decomposed, and their stability is lower than that of dry MOF-5. Studies show that the contact of a large number of water molecules can densify the surface of MOF-5, thus slowing down the further hydrolysis of MOF-5, but can not prevent its decomposition.(Ming Y , Purewal J et al. Kinetic Stability of MOF-5 in Humid Environments: Impact of Powder Densification, Humidity Level, and Exposure Time[J]. Langmuir, 2015, 31:4988-4995.)In clinical application, this characteristic has the advantage of reducing the accumulation in vivo. Because there are many studies on water stability of MOF-5, this paper has not carried out such an investigation.

Thank you again for your patient review and valuable advice.

Round 2

Reviewer 1 Report

The revised version of the manuscript is suitable for publication.

Author Response

Dear expert reviewer, thank you for your guidance and suggestions.

Reviewer 2 Report

I am very disappointed with the answers, and I challenge the authors to write a convincing rebuttal.

Authors mentioned in the abstract" The results showed that nano-MOF-5 prepared by direct addition method had complete structure, uniform size, and good biocompatibility and was suitable as ORI carrier"

a) Is it true that these materials have nano-size? b) I persistently ask you to measure crystallites on the basis of the presented SEM images.

Authors wrote in answer 3 " maximum loading value of 1.11 g (3.04mmol) of ORI per gram of MOF-5" Is this data available in new the version of the manuscript?

I am not entirely convinced about the integrity of the MOF-5 and ORI@MOF-5 materials.

a)The IR and XPRD data before and after loading ORI in MOF-5 have to be added and this needs to be discussed and rationalized in the new version manuscript.

b)Why were MOF-5 and ORI@MOF-5 N2 isotherm / BET surface area and porosity data omitted?

Author Response

Point 1: Is it true that these materials have nano-size? I persistently ask you to measure crystallites on the basis of the presented SEM images

Response 1: Dear expert reviewer, MOF-5 synthesized by direct addition method does have nano-size. We have verified it by DLS, as shown in Fig. 5. In addition, we have listened to your opinions and measured the size of microcrystals on the SEM diagram, as shown in Figure 4.

Point 2: Authors wrote in answer 3 " maximum loading value of 1.11 g (3.04mmol) of ORI per gram of MOF-5" Is this data available in new the version of the manuscript?

Response 2: Dear expert reviewer, this representation is not used in the article. I only describe it in the reversion document.

The maximum drug loading was 52.58% in this study by using the following equation:

Loading capacity=(MO-MuO)/(MO-MuO+MM )×100%  

MO, MuO, and MM are the total amount of ORI, the un-loaded amount of ORI in the supernatant, and the amount of MOF-5, respectively.

Through formula conversion, it can also be expressed as 1.11 g (3.04mmol) of ORI per gram of MOF-5.

I have added this data to the article, see 3.2. Drug loading.

Point 3: The IR and XPRD data before and after loading ORI in MOF-5 have to be added and this needs to be discussed and rationalized in the new version manuscript

Response 3: Dear expert reviewer, we have taken your advice and added the IR and XRD data of MOF-5 and ORI@MOF-5. And that was discussed in the new version manuscript.

Point 4: Why were MOF-5 and ORI@MOF-5 N2 isotherm / BET surface area and porosity data omitted?

Response 4: Dear expert reviewer, thank you for your comments. We have added N2 isotherm / BET surface area and porosity data in the article. BET analysis showed that the specific surface area and pore volume of the material were significantly reduced after drug loading, indicating that the drug was successfully loaded.